# A Novel Single Tube Semi-Active Tuned Liquid Gas Damper for Suppressing Horizontal Vibrations of Tower-like Structures

**Michael Reiterer** [1,*] **and Janez Schellander** [2]

1    Institute of Structural Dynamics, Vienna University of Technology, 1040 Vienna, Austria
2    Department of Structural Dynamics, REVOTEC Engineering, 1070 Vienna, Austria;
     janez.schellander@revotec.at
*    Correspondence: michael.reiterer@revotec.at; Tel.: +43-660-2999-363

**Abstract:** The purpose of this paper is to present a novel single tube semi-active tuned liquid gas damper (SA-TLGD) for suppressing horizontal vibrations of tower-like structures and to study its damping effectiveness. The main difference to the well-known state-of-the-art tuned liquid column damper (TLCD) is the special geometric shape of the developed SA-TLGD. Contrary to the TLCD, the presented SA-TLGD only consists of a single horizontal tube that is partially filled with water. A large deformable elastic membrane with neglectable stiffness is used as the interface between the liquid and the air. Both ends of the horizontal tube are sealed and the resulting gas spring is used as the restoring force and frequency tuning parameter, respectively. The developed SA-TLGD is a semi-active vibration damping device, where its natural frequency and magnitude of energy dissipation can be re-adjusted during operation. Due to the lack of any vertical tube parts, this new type of vibration absorber requires significantly less installation space compared to the classical TLCDs. The equations of motion of the SA-TLGD and the coupled main system are derived by the application of conservation of momentum. The procedure of optimal tuning of the SA-TLGD is presented, and computational numerical studies are performed to demonstrate the damper effectiveness. It is shown that the application of the developed SA-TLGD provides a large reduction in the maximum horizontal forced vibration amplitudes of tower like-structures and that its semi-active functionality enables the possibility of re-adjustment any time during the operation life of the structure.

**Keywords:** tower-like structures; tuned liquid gas damper; semi-active; elastic membrane; gas spring; gas volume; SA-TLGD; TLCGD; TLCD; TMD





## 1. Introduction

Tower-like structures, e.g., high-rise buildings, chimneys, and wind turbines, are prone to vibrations when subjected to wind, sea waves, and earthquake loads, which may cause structural failure, discomfort to occupants, and malfunction of the installed equipment. Hence, the mitigation of structural vibrations has always been a major concern amongst structural engineers. One of the effective means to reduce the dynamic response of tower-like structures is the application of dynamic vibration absorbers. The Tuned Mass Damper (TMD) is one of the most popular passive control systems and has been broadly studied and applied to many engineering structures [1–9]. TMDs make use of a moving secondary mass capable of counteracting the dynamic motion of the vibrating structure.

Amongst the widely recognized application of TMDs, Tuned Liquid Dampers (TLDs) have also become very popular for vibration suppression of tower-like structures in recent decades. In civil engineering applications, the following two types of TLDs are commonly utilized: Tuned Sloshing Dampers (TSDs) and Tuned Liquid Column Dampers (TLCDs). TSDs are typically constructed by a tank partially filled with water and they can be either based upon a deep or a shallow water configuration. TSDs absorb and dissipate kinetic energy through boundary layer friction, wave breaking, and free surface rupture during the

interaction with the vibrating structure [10–14]. Pandey et al. [15] studied a compliant tuned liquid damper for controlling seismic vibrations of short period structures by mounting a TLD on an array of compliant elastomeric pads and in [16] is the authors propose to implement a tuned liquid mass damper in a deep liquid storage tank by flexibly attaching the tank to the structure to allow tuning of vibration of the impulsive mass to short period structures.

TLCDs are a special type of TLDs that rely on the motion of a liquid mass in a rigid U-shaped tube to counteract the action of external forces acting on the structure, with the inherent damping being introduced in the oscillating liquid column through friction and built-in orifices. The tank consists of two vertical columns and a single horizontal connecting tank. TLCDs are partially filled with a Newtonian fluid until the liquid reaches a certain level in the two vertical columns. Due to its cost-effectiveness, simplicity in installation, and low maintenance costs, TLCDs have attracted significant interest for researchers and engineers [17–21]. The applicability of passive TLCDs is limited to civil engineering structures with extremely low vibrating frequencies up to around 0.5 Hz. However, this detrimental property of TLCDs is conquered through some innovative ideas such as utilizing an air spring (gas spring) in the vertical tubes, to extend the range of applicability to structural vibration frequencies up to 5.0 Hz [22,23]. The resulting beneficial frequency increase and tuning effects due to the gas spring are reflected by the so-called Tuned Liquid Column Gas Damper (TLCGD). The application of TLCGDs offers a quite simple tuning mechanism since the natural frequency can be adjusted by pressurizing the gas chambers or adjusting the size of the gas volume inside the vertical sealed tubes [24,25]. Further developments of TLDs and TLCDs were studied by Zhao et al. [26] who presented a novel tuned liquid inerter system by employing the synergy benefits of an inerter-based subsystem and a tuned liquid element to achieve the lightweight-based improved control performance and by Di Matteo et al. [27] who studied a tuned liquid column damper inerter to control the seismic response of structural systems.

For a passive vibration absorber, designed with optimal tuned frequency and damping ratio, these optimum parameters are valid only for a given level of wind, sea waves, or seismic excitation with a specific frequency content. In fact, wind, sea waves, and earthquake forces acting on tower-like structures are random in nature, with their extent and frequency content being different at different times. Likewise, the dynamic properties of a tall building structure, including the natural frequencies and damping ratios, are response-amplitude-dependent during strong winds [28]. It is therefore highly desirable, to develop frequency- and damping-variable or parameter-adjustable vibration absorbers to achieve optimal control performance for a wide range of loading conditions and therefore to be able to consider structural uncertainties [29,30]. Structures also show a significant change in their natural frequencies and damping ratios with increasing age and due to temperature effects. Hence, Yalla et al. [31] proposed a semi-active TLCD which achieves variable fluid damping by using a controllable valve to adjust the orifice opening and Haroun et al. [32] presented a concept of a hybrid liquid column damper that can actively control the orifice opening ratio. Altay and Klinkel [33] presented a semi-active TLCD that provides mechanisms for a continuous adaptation of both its natural frequency and damping behavior in real time. Further relevant research in the field of semi-active TLCDs has been performed by Wang et al. [34] and Sarkar and Chakraborty [35]. They presented a semi-active TLCD with the use of magneto-rheological (MR) fluids to generate controllable fluid damping. The MR fluids are smart materials that can reversibly change from a free-flowing, linear viscous fluid to a semi-solid with a controllable yield strength in milliseconds when exposed to a magnetic field [36]. Thus, they are used as damping fluids to devise semi-active MR-TLCDs with alterable fluid viscosity. The strongly modifiable fluid viscosity results in adjustable and controllable damping forces in the MR-TLCD for structural vibration control under a wide range of loading conditions.

Regarding existing installations of TLDs onto civil engineering structures, their sizes can vary, depending on the kinetic equivalent moving mass of the structure, from relatively compact units to much larger devices of several hundred tonnes (e.g., the two TLCDs

installed at the top of the 52-storey Random House Tower in New York City of, respectively, 265,000 and 379,000 kg [37]). In the case of TLDs, the apparent low density of the moving damper mass (e.g., water $\rho$ = 1000 kg/m$^3$) compared to classical tuned mass dampers (e.g., steel $\rho$ = 7850 kg/m$^3$) results in a considerable disadvantage in terms of the space required at the installation site of the damper. Especially slender vibration-prone structures, e.g., wind turbines, only provide a very small installation space at the tower head and, hence, the practical application of TLDs is a quite challenging task. For instance, a 5 MW offshore wind turbine with a 112 m steel tube length has a diameter of just 5.5 m at the tower head [38]. To achieve the desired bidirectional damping effect, at least two TLDs must be installed in the two relevant vibration directions, resulting in a very large space requirement at the tower head. In the case of installation of TLCDs, the two vertical columns of the U-shaped tank, which are arranged at a distance from each other and communicate through the horizontal passage, also need an enormous installation space inside of the structure. Thus, it is evident that although TLDs offer several advantages compared to TMDs, their application to real structures often fails because of the limited available installation space in the case of vibration-prone slender tower-like structures, e.g., wind turbines. Compared to the liquid dampers, the installation space required for the moving mass of pendulum dampers is significantly smaller due to the density of steel. However, the very low fundamental natural frequency in the range of approx. 0.20 Hz usually found in wind turbines and high-rise structures requires a very large pendulum length of approx. 6.21 m. Hence, the space gained by the small steel pendulum mass compared to the fluid mass of liquid dampers is compensated by the very long pendulum lengths.

In this paper, a novel single tube semi-active tuned liquid gas damper (SA-TLGD) for suppressing horizontal vibrations of tower-like structures is presented and its damping effectiveness is studied computationally considering a SDOF- and MDOF-main system. The novelty of the damper lies in the lack of any vertical columns and the design of a single tube only which is partially filled with a fluid, e.g., water (Figure 1). A large deformable elastic membrane with neglectable stiffness is used as an interface between the fluid and the air. The SA-TLGD can be interpreted as a TLCGD, but without any vertical columns, i.e., the restoring force from gravity is not present anymore. Both ends of the horizontally orientated tube are sealed and the resulting gas spring is used as the restoring force and frequency tuning parameter, respectively. To adjust the SA-TLGD vibrating frequency, the bulk gas volume $V_0$ is separated into a series of gas chambers $V_{0i}$ that are connected via controllable valves. Depending on the desired optimal vibrating frequency, a specific size of gas volume is initiated through the utilized control software, running on the microcontroller (MC), which opens or closes the appropriate number of valves. It is noted that the valves are either in a completely closed or opened position, i.e., they do not work as damping-increasing throttling valves. The magnitude of the fluid damping is properly adjusted by varying the diameter of several controllable orifices.

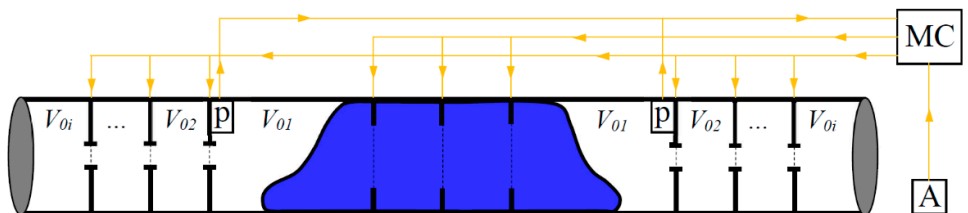

**Figure 1.** Novel single tube semi-active Tuned Liquid Gas Damper (SA-TLGD).

The semi-active control logic of the SA-TLGD is based on measuring the present resonance frequency of the vibration-prone main system with properly installed accelerometers (A). Hence, the input parameter for the control algorithm is the actual resonance frequency calculated from the recorded acceleration response in the time domain by application of FFT. Based on this resonance frequency information, the microcontroller (MC) sets the optimal number of opened or closed valves and the optimal diameter of the orifices to

achieve the optimal absorber parameters. The actual vibration response and vibrating frequency, respectively, of the fluid mass is measured with pressure sensors (p). Hence, the developed SA-TLGD is a semi-active vibration damping device, where its natural frequency and magnitude of energy dissipation can be re-adjusted during the operation of the damper at any time. However, the adjustment of the damper parameters to the optimal values is not performed in real time immediately after detecting slight changes in the main system resonance frequency, but after detecting significant changes during the structural lifetime and, therefore, it is not necessary to adjust the damper parameters in real time during the state of fluid mass vibration. Due to the lack of any vertical tubes, this new type of vibration absorber requires significantly less installation space compared to classical TLCDs and its semi-active functionality ensures an optimal performance over the total operating life of vibration-prone structures. It is noted that the studied SA-TLGD has no fixed direction and, hence, it can act in arbitrary planes. Regarding the long-term durability of the SA-TLGD, it is important that all construction elements of the damper are designed for both the static and dynamic loading conditions that occur during the total operating lifetime.

## 2. Mechanical Model

Tower-like structures with low inherent damping are, in general, forced to couple bending and torsional vibrations. In this study, it is assumed that the vibration modes of the considered structure are well separated and, hence, modal tuning of SA-TLGD is applicable to a selected vibration mode. The mechanical model is developed in steps, starting with the free body diagram of the SA-TLGD (Figure 2a) and the formulation of in-plane rigid body motions of a seismic $w_g(t)$ and force $F(t)$ excited SDOF-shear frame structure with a single SA-TLGD attached (Figure 2b). The substructure synthesis method is applied to derive the coupled equations of motion.

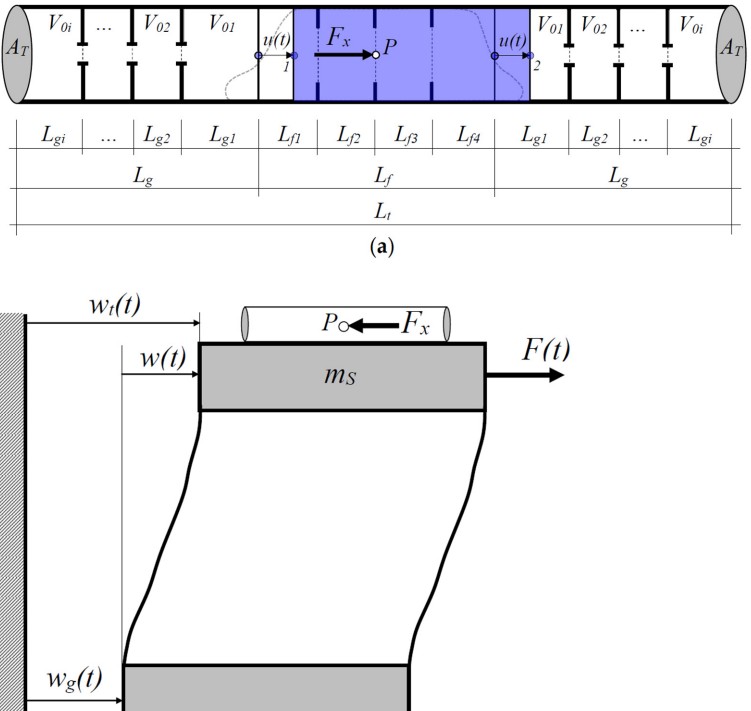

**Figure 2.** Free body diagrams of the mechanical model: (**a**) single tube SA-TLGD under horizontal floor excitation; (**b**) SDOF-main system under combined seismic $w_g(t)$ and force excitation $F(t)$ with horizontal interaction force $F_x$ from the SA-TLGD dynamics.

*2.1. Free Body Diagram of SA-TLGD*

　　The single tube SA-TLGD is considered in a deflected position separated from the floor of the SDOF-main system under horizontal floor excitations $w_t = w_g + w$ (Figure 2). The horizontal orientated tube with diameter $A_T$ is sealed at both ends and filled with a fluid (density $\rho_f$) over the length $L_f$, i.e., the total fluid mass yields to $m_f = \rho_f A_T L_f$. The relative motion of the fluid interface to gas is described by the displacement $u_1 = u_2 = u(t)$. A large deformable elastomeric membrane with neglectable stiffness is used as the interface. To formulate the SA-TLGDs dynamic, the shape of the membrane in static equilibrium is approximated as a pure vertically orientated surface. During the vibration process, the gas inside the air chambers $V_{0i}$ is quasi-statically compressed and released by the moving fluid interface in relatively slow motion. A pressure difference $\Delta p = p_2 - p_1$ is created between the left and right gas chambers and it changes the undamped circular natural frequency of the SA-TLGD defined in Equation (8). The absolute acceleration of the chosen reference Point $P$ in the prescribed rigid body motion is given by

$$\ddot{w}_t = \ddot{w}_g + \ddot{w}. \tag{1}$$

　　To derive the nonlinear equation of motion, the floor excited SA-TLGD is transformed into an equivalent single-mass oscillator [39]. Thereby, the fluid mass $m_f = \rho_f A_T L_f$ represents the equivalent mass $M^*$ of the harmonic oscillator. Neglecting the stiffness contribution of the elastic membranes, the equivalent spring stiffness $K^*$ results solely from the gas spring effect. It is assumed that the gas pressure $p_0$ is present in both gas chambers in the static equilibrium position of the fluid. $p_0$ can either be chosen equal to the atmospheric pressure ($p_0 \cong 10^5$ Pa) or to any other arbitrary desired value (negative or positive pressure). The initial size of the gas volume $V_0$ enclosed on the left and right sides of the fluid in the static equilibrium position is exposed to a constant change in compression and expansion, i.e., the size of $V_0$ fluctuates as a function of the horizontal dynamic deformation $u(t)$ of the fluid mass. Hence, the time varying size of $V_0$ leads to a change in the pressure state of the gas volume and in consequence of a restoring effect of the deflected fluid mass. The resulting gas pressure difference $\Delta p = p_2 - p_1$ approximately follows the quasistatic polytropic law [39] ($\rho_0$ is the gas density in the static equilibrium position of the fluid mass),

$$\Delta p = p_0 \left( \frac{\Delta \rho}{\rho_0} \right)^n = p_0 \left( \frac{V_0}{\Delta V} \right)^n, \ \Delta V = V_2 - V_1, \ V_0 = \sum_{i=1}^{k} V_i. \tag{2}$$

where $n$ denotes the polytropic coefficient and $k$ is the number of conducted separated gas chambers $V_i$. In the case of very slow fluid velocities (low eigenfrequency) the gas spring acts approximately under isothermal conditions $n = 1.0$, while in the case of higher velocities an adiabatic condition occurs and the coefficient changes to a value of $n = 1.4$. In any other circumstances, $n$ takes a value in the range between 1.0 and 1.4 [25]. During the state of fluid mass vibration, the actual size of gas volume at the left and right gas chambers is defined as (Figure 2a),

$$V_1 = V_0 + A_T u, \ V_2 = V_0 - A_T u. \tag{3}$$

　　Inserting the expressions for the gas volumes $V_1$ and $V_2$ into Equation (2) yields the nonlinear gas pressure difference,

$$\Delta p(u) = p_0 \left[ \left( \frac{V_0}{V_0 - A_T u} \right)^n - \left( \frac{V_0}{V_0 + A_T u} \right)^n \right]. \tag{4}$$

A linearization of Equation (4) is obtained using a Taylor series expansion of the nonlinear function with respect to the equilibrium pressure $p_0$ while neglecting the higher order terms,

$$\Delta p(u) = \frac{2\,n\,p_0\,A_T}{V_0}\,u + O\left(u^3\right) \approx \frac{2\,n\,p_0}{L_g}\,u,\tag{5}$$

where $L_g = V_0/A_T$ defines the horizontal length of the gas spring at the left and right gas chambers of the SA-TLGD (Figure 2a). From Equation (5) and the relation $K^*\,u = \Delta p\,A_T$ the single-mass oscillator equivalent spring stiffness $K^*$ results as

$$K^* = \frac{2\,n\,p_0\,A_T}{L_g}.\tag{6}$$

The deviation of the exact solution for the nonlinear relative pressure difference $\Delta p/p_0$ (Equation (4)) from the linearized solution (Equation (5)) is illustrated in Figure 3 as a function of the dimensionless relative displacement $u(t)/L_g$. The polytropic coefficient was chosen with $n = 1.2$, i.e., mean value of isotherm and adiabatic state change. It is indicated that up to $u(t)/L_g \leq 0.30$ the deviation to the exact nonlinear solution is insignificantly small, i.e., linearization is permissible in this range of the vibration amplitude. In the case of very large vibration amplitudes $u(t)/L_g > 0.50$, the restoring force of the nonlinear gas spring differs significantly from the linear solution and, therefore, leads to a disadvantageous detuning of the SA-TLGD eigenfrequency. Hence, when designing the SA-TLGD the relation of the maximum fluid mass displacement $u(t)$ to the chosen gas spring length $L_g$ is an important tuning parameter to achieve the optimum vibration damping effect. For this reason, the maximum amplitude of gas compression is limited to $max\,|u(t)| = U_{max} < 0.30\,L_g$ to keep the eigenfrequency of the SA-TLGD approximately constant.

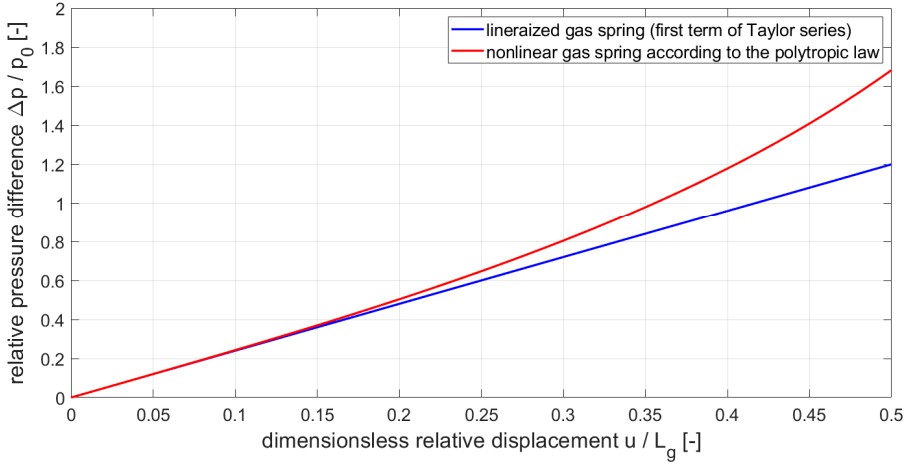

**Figure 3.** Comparison of linear and nonlinear gas spring stiffness function $\Delta p/p_0$.

After derivation of the equivalent mass $M^* = m_f = \rho_f\,A_T\,L_f$ and stiffness $K^*$ given in Equation (6) of a single-mass oscillator, the nonlinear equation of motion for the floor excited SA-TLGD yields to,

$$\ddot{u} + \delta_L|\dot{u}|\,\dot{u} + \omega_A^2\,u = -\ddot{w}_t,\tag{7}$$

where an averaged turbulent damping term, that must be experimentally verified, has been added. The head loss coefficient $\delta_L$ can be increased by properly selecting the diameter of the built-in orifice plate (Figure 2a). In the case of stationary flow, $\delta_L$ is tabulated for

relevant pipe elements and cross-sectional areas in [40]. In Equation (7), the undamped circular natural frequency $\omega_A$ of the both side sealed SA-TLGD is defined as,

$$\omega_A = \sqrt{\frac{2\,n\,p_0\,A_T}{m_f\,L_g}}\,. \tag{8}$$

In Equation (8) $n$, $p_0$, $A_T$, $m_f$, and $L_g$ denote the polytropic coefficient ($1.0 \leq n \leq 1.4$), the initial equilibrium pressure in the gas chambers, the cross-sectional area of the tube, the fluid mass, and the total gas spring length. $L_g = V_0/A_T$ is an important design variable of the SA-TLGD, defining the volume of the gas chamber in terms of the cross-sectional area. The re-optimization of the damper eigenfrequency is performed during the state of operation by semi-actively controlling the appropriate number of either closed or open valves, i.e., adjusting the size of the gas volume $V_0$. Equation (8) rewritten into a more general form leads to,

$$\omega_A = \sqrt{\frac{2\,n\,p_0\,A_T^2}{m_f\,\sum\limits_{i=1}^{k} V_{0i}}}\,. \tag{9}$$

Linearization of the Fluid Flow Equation of Motion

Regarding the application of the absorber optimization procedure in Section 3, the nonlinear turbulent damping term $\delta_L|\dot{u}|\dot{u}$ in Equation (7) must be transformed into its equivalent linear one, $2\zeta_A\omega_A\dot{u}$. Demanding equally dissipated energy during one cycle, over a vibration period $T$, for the nonlinear and the linear SA-TLGD results in the relation,

$$\int_0^T \left|\left(\delta_L|\dot{u}|\,\dot{u} + \omega_A^2\,u\right)\dot{u}\right| dt = \int_0^T \left|\left(2\,\zeta_A\,\omega_A\,\dot{u} + \omega_A^2\,u\right)\dot{u}\right| dt, \tag{10}$$

and, when substituting the time harmonic displacement function, $u(t) = U_0\,\cos\omega_A t$, the equivalent linear viscous damping coefficient can be written proportional to the absorber vibration amplitude as follows,

$$\zeta_A = \frac{4\,U_0\,\delta_L}{3\,\pi}. \tag{11}$$

Under these conditions, Equation (7) takes on its linearized form with,

$$\ddot{u} + 2\,\zeta_A\,\omega_A\,\dot{u} + \omega_A^2\,u = -\ddot{w}_t, \; |u| \leq U_0 \approx U_{max}. \tag{12}$$

The value for $U_0$, used in Equation (11) for general forced vibrations of the fluid mass, is determined by means of numerical simulations of the linear coupled main system with the SA-TLGD attached and commonly chosen as $U_0 = U_{max}$.

Applying the conservation of momentum to the fluid mass in the free body diagram of the SA-TLGD (Figure 2a) determines the resultant horizontal interaction force,

$$F_x = m_f\left(\ddot{w}_t + \ddot{u}\right), \; m_f = \rho_f A_T L_f. \tag{13}$$

*2.2. Substructure Synthesis of a SDOF-Main System with SA-TLGD Attached*

The SDOF-main system with the assigned interaction force $F_x$ from the SA-TLGD dynamics under combined seismic $\ddot{w}_g$ and force excitation $F(t)$, is considered in the next step of the substructure synthesis method (Figure 2b). The external dynamic force $F(t)$ is interpreted as a wind load. The main system floor deformation is given by the displacement $w(t)$, whereby the time variant P-Δ effect is neglected. The moving floor mass $m_H$ includes the dead weight $m_{D,A}$ of the SA-TLGD and the modal masses of the vertical columns. The field stiffness of the columns is denoted with $k$, and it includes the geometric correction

of prestressing by the dead weight considered for the columns according to [39,41]. Light structural damping is assumed by the damping coefficient $b$.

Conservation of momentum of the floor mass $m_S$ yields the relevant linear equation of motion of the SDOF-main system with the attached SA-TLGD (Figure 2b),

$$\ddot{w} + 2\,\zeta_S\,\Omega_S\,\dot{w} + \Omega_S^2\,w = -\ddot{w}_g + \frac{F(t)}{m_S} - \frac{F_x}{m_S}, \quad \zeta_S = \frac{b}{2\,m_S\,\Omega_S}, \quad \Omega_S^2 = \frac{k_S}{m_S}, \quad (14)$$

where $\Omega_S$ and $\zeta_S$ represent the undamped circular natural frequency of the SDOF-main system, and the linear viscous damping ratio. Inserting the coupling force $F_x$ from Equation (13) into Equation (14) leads together with Equation (12) to the coupled system of linearized equations of motion for the seismic and force excited 2-DOF system,

$$\begin{array}{l}\ddot{w} + 2\,\zeta_S\,\Omega_S\,\dot{w} + \Omega_S^2\,w = -\ddot{w}_g + \frac{F(t)}{m_S} - \mu\,\ddot{w}_t - \mu\,\ddot{u}\,,\\[4pt]\ddot{u} + 2\,\zeta_A\,\omega_A\,\dot{u} + \omega_A^2\,u = -\ddot{w}_t\,,\\[6pt]w_t = w_g + w\,.\end{array} \quad (15)$$

The ratio of fluid mass to the moving mass of the SDOF-main system is defined by,

$$\mu = \frac{m_f}{m_S}. \quad (16)$$

To provide highest possible vibration suppression of the main system the mass ratio $\mu$ should be maximized. However, from a practical point of view, i.e., limited installation space and avoidance of detrimental frequency shifts of the main system, the mass ratio $\mu$ is typically chosen in range of 0.5–5%.

To prepare for the equations of motion of a MDOF-main system with multiple, differently tuned SA-TLGDs attached, Equation (15) is rewritten in its linearized matrix form,

$$\begin{array}{l}\mathbf{M_S}\begin{bmatrix}\ddot{w}\\\ddot{u}\end{bmatrix} + \mathbf{C_S}\begin{bmatrix}\dot{w}\\\dot{u}\end{bmatrix} + \mathbf{K_S}\begin{bmatrix}w\\u\end{bmatrix} = -\begin{bmatrix}m_S + m_f\\1\end{bmatrix}\ddot{w}_g + \begin{bmatrix}F(t)\\0\end{bmatrix},\\[12pt]\mathbf{M_S} = \begin{bmatrix}m_S + m_f & m_f\\1 & 1\end{bmatrix}, \quad \mathbf{C_S} = \begin{bmatrix}2\zeta_S\Omega_S m_S & 0\\0 & 2\zeta_A\omega_A\end{bmatrix}, \quad \mathbf{K_S} = \begin{bmatrix}k_S & 0\\0 & \omega_A^2\end{bmatrix}.\end{array} \quad (17)$$

### 2.3. Substructure Synthesis of a MDOF-Main System with Multiple SA-TLGD Attached

Based on Equation (17), the coupled linearized equations of motion of a seismic and forced excited MDOF-main system with multiple SA-TLGDs attached are described by the following set of matrix equations, in a hyper matrix formulation,

$$\begin{array}{l}\mathbf{M_S}\begin{bmatrix}\ddot{\vec{w}}\\\ddot{\vec{u}}\end{bmatrix} + \begin{bmatrix}\mathbf{C} & 0\\0 & \mathbf{C_f}\end{bmatrix}\begin{bmatrix}\dot{\vec{w}}\\\dot{\vec{u}}\end{bmatrix} + \begin{bmatrix}\mathbf{K} & 0\\0 & \mathbf{K_f}\end{bmatrix}\begin{bmatrix}\vec{w}\\\vec{u}\end{bmatrix}\\[14pt]= -\begin{bmatrix}\mathbf{M}\,\vec{r}_S + \mathbf{L}\,\mathbf{M_f}\,\vec{i}\\\vec{i}\end{bmatrix}\ddot{w}_g + \begin{bmatrix}\vec{F}(t)\\0\end{bmatrix}.\end{array} \quad (18)$$

where $\vec{r}_S$ denotes the static influence vector, which for the single point base excitation renders to $\vec{r}_S = \vec{i} = [1\,1\,\ldots\,1]^T$. In Equation (18) the sparse SA-TLGD position matrix

**L** with dimension $N \times n_A$, where $N$ and $n_A$ define the main system degree of freedom (DOF) and the number of installed SA-TLGDs,

$$
\mathbf{L} = \begin{bmatrix} 1 & 0 & 1 \\ \vdots & \vdots & \vdots \\ 0 & 1 & 0 \\ \vdots & \vdots & \vdots \\ 0 & 0 & 0 \end{bmatrix} \quad \leftarrow \; DOF \; to \; be \; influenced, \; N
$$

$$
\underset{number \; of \; SA-TLGD, \; n_A}{\uparrow}
$$

(19)

is included into the generalized mass matrix $\mathbf{M_S}$ as well,

$$
\mathbf{M_S} = \begin{bmatrix} \mathbf{M} + \mathbf{L}\,\mathbf{M_f}\,\mathbf{L^T} & \mathbf{L}\,\mathbf{M_f} \\ \mathbf{L} & \mathbf{I} \end{bmatrix}.
$$

(20)

**M**, **C**, and **K** are the mass, damping, and stiffness matrices of the MDOF-main system and the SA-TLGD parameter-related diagonal matrices $\mathbf{M_f}$, $\mathbf{C_f}$, and $\mathbf{K_f}$ are defined as follows,

$$
\mathbf{M_f} = diag \left[ m_{f1}, \; \ldots \ldots, \; m_{fn_A} \right],
$$

$$
\mathbf{C_f} = diag \left[ 2\zeta_{A1}\omega_{A1}, \; \ldots \ldots, \; 2\zeta_{An_A}\omega_{An_A} \right],
$$

$$
\mathbf{K_f} = diag \left[ \omega_{A1}^2, \; \ldots \ldots, \; \omega_{An_A}^2 \right].
$$

(21)

## 3. Optimal Tuning of SA-TLGDs

### 3.1. Optimal Tuning of a Single SA-TLGD Attached to a SDOF-Main System

For optimal modal tuning of a single SA-TLGD attached to a SDOF-main system, the two design parameters $\delta = \omega_A / \omega_S$ and $\zeta_A$ must be selected appropriately. Here, $\delta$ defines the ratio between the natural frequency of the vibration absorber and the natural frequency of the SDOF-main system, and $\zeta_A$ defines the linearized viscous damping ratio of the SA-TLGD. The analytical formulas for determining the optimal design parameters for the classical TMD attached to a SDOF-main system were first presented by Den Hartog [1], for a time harmonic force excitation and an undamped main system $\zeta_S = 0$,

$$
\delta_{opt} = \frac{\omega_A}{\omega_S} = \frac{1}{1+\mu}, \; \zeta_{A,opt} = \sqrt{\frac{3\mu}{8\,(1+\mu)}}, \; \mu = \frac{m_f}{m_S}.
$$

(22)

Equation (22) is applicable for determining the optimal design parameters to minimize the displacement $w(t)$ of a harmonic force excited main system $F(t) = F_0 \, e^{ivt}$, $v$ is the excitation frequency, and it remains unchanged when minimizing the acceleration $\ddot{w}(t)$ of a harmonic base excited main system [7]. Slightly different parameters result for a harmonic base excited main system $w_g(t) = w_0 \, e^{ivt}$ when minimizing the displacement $w(t)$ [7],

$$
\delta_{opt} = \sqrt{\frac{2-\mu}{2\,(1+\mu)^2}}, \; \zeta_{A,opt} = \sqrt{\frac{3\mu}{4\,(1+\mu)\,(2-\mu)}}.
$$

(23)

The Equations (22) and (23) indicate that the optimum design parameters of the single SA-TLGD attached to a SDOF-main system depend solely on the mass ratio $\mu = m_A / m_S$. In practical applications, the mass ratio is usually chosen in range of 0.5–5%. It is noted that the optimal design parameters given in Equations (22) and (23) are derived under the assumption of an undamped main system, i.e., the optimal performance of the vibration absorber is present for $\zeta_S = 0$. However, Pocanschi and Phocas [42] give the following

analytical equations for the correction of the optimal design parameters considering the structural damping of the main system $\zeta_S > 0$ when optimizing the vibration damper,

$$
\begin{aligned}
\widetilde{\delta}_{opt} &= \delta_{opt} - \left(0,241 + 1,7\,\mu - 2,6\,\mu^2\right)\zeta_S - \left(1 - 1,9\,\mu + \mu^2\right)\zeta_S^2, \\
\widetilde{\zeta}_{A,opt} &= \zeta_{A,opt} + \left(0,13 + 0,12\,\mu + 0,4\,\mu^2\right)\zeta_S - \left(0,01 + 0,9\,\mu + 3\,\mu^2\right)\zeta_S^2.
\end{aligned}
\tag{24}
$$

### 3.2. State Space Optimal Tuning of Multiple SA-TLGDs Attached to a MDOF-Main System

In the case of multiple SA-TLGD attached to a MDOF-system, the tuning process is best performed in two steps. At first, the linearized model is tuned with respect to a selected vibration mode of the main system using the classical Den Hartog formulas, presented in Section 3.1. Fine tuning of the absorber parameters for MDOF-systems is best achieved by also considering adjacent vibration modes in a state space representation, by minimizing the weighted squared area of the frequency response function (FRF). Hence, the coupled linearized equations of motion of the seismic and force excited MDOF-main system with multiple SA-TLGDs attached (Equation (18)), are transformed into the state space. Introducing the state space hypervector $\vec{z} = \left[\vec{w}\ \vec{u}\ \dot{\vec{w}}\ \dot{\vec{u}}\right]$ and its time derivative $\dot{\vec{z}}$, renders the first order matrix equation in state space,

$$
\begin{aligned}
\dot{\vec{z}} &= (\mathbf{A} + \mathbf{B}\,\mathbf{R})\,\vec{z} - \vec{e}_g\ddot{w}_g + \vec{b}, \\
\vec{e}_g &= \left[\vec{0}\quad \vec{0}\quad \mathbf{M_S^{-1}}\left(\begin{array}{c} \mathbf{M}\,\vec{r}_S + \mathbf{L}\,\mathbf{M_f}\,\vec{i} \\ \vec{i} \end{array}\right)\right], \\
\vec{b} &= \mathbf{E_F}\,\vec{F}, \quad \mathbf{E_F} = \left[0\quad 0\quad \mathbf{M_S^{-1}}\left(\begin{array}{c} \mathbf{I} \\ \mathbf{0} \end{array}\right)\right].
\end{aligned}
\tag{25}
$$

The forcing hypervector contains the horizontal excitation forces $\vec{F}(t)$. Since only the matrix elements $\mathbf{A}$ and $\mathbf{B}$ from Equation (25) contain the known main system parameters, the state space matrix $\mathbf{A} + \mathbf{B}\,\mathbf{R}$ should be kept separated,

$$
\mathbf{A} = \left[\begin{array}{cccc}
\mathbf{0} & \mathbf{0} & \mathbf{I} & \mathbf{0} \\
\mathbf{0} & \mathbf{0} & \mathbf{0} & \mathbf{I} \\
-\mathbf{M_S^{-1}}\left(\begin{array}{cc}\mathbf{K} & \mathbf{0}\\ \mathbf{0} & \mathbf{0}\end{array}\right) & & -\mathbf{M_S^{-1}}\left(\begin{array}{cc}\mathbf{C} & \mathbf{0}\\ \mathbf{0} & \mathbf{0}\end{array}\right) &
\end{array}\right],
$$

$$
\mathbf{B} = \left[\begin{array}{cccc}
\mathbf{0} & \mathbf{0} & \mathbf{I} & \mathbf{0} \\
\mathbf{0} & \mathbf{0} & \mathbf{0} & \mathbf{I} \\
-\mathbf{M_S^{-1}}\left(\begin{array}{cc}\mathbf{I} & \mathbf{0}\\ \mathbf{0} & \mathbf{I}\end{array}\right) & & -\mathbf{M_S^{-1}}\left(\begin{array}{cc}\mathbf{I} & \mathbf{0}\\ \mathbf{0} & \mathbf{I}\end{array}\right) &
\end{array}\right],
\tag{26}
$$

and the matrix $\mathbf{R}$ contains the unknown SA-TLGD design parameters,

$$
\mathbf{R} = \left[\begin{array}{cccc}
\mathbf{0} & \mathbf{0} & \mathbf{0} & \mathbf{0} \\
\mathbf{0} & \mathbf{K_f} & \mathbf{0} & \mathbf{0} \\
\mathbf{0} & \mathbf{0} & \mathbf{0} & \mathbf{0} \\
\mathbf{0} & \mathbf{0} & \mathbf{0} & \mathbf{C_f}
\end{array}\right].
\tag{27}
$$

For the seismic excited MDOF-main system with multiple SA-TLGDs attached, the steady-state solution in the frequency domain results as,

$$
\vec{z}(\nu) = \left[i\nu\,\mathbf{I} - (\mathbf{A} + \mathbf{B}\,\mathbf{R})\right]^{-1}\vec{e}_g.
\tag{28}
$$

For the force excited coupled MDOF-main system the steady state solution is given by,

$$
\vec{z}(\nu) = \left[i\nu\,\mathbf{I} - (\mathbf{A} + \mathbf{B}\,\mathbf{R})\right]^{-1}\vec{b}_0.
\tag{29}
$$

To determine the optimal design parameters of the SA-TLGD it is common practice to minimize an appropriate performance index, e.g., defined by the infinite integral of the weighted sum of quadratic state variables of the MDOF-main system $\vec{z}_S$, in the frequency domain, see, e.g., Müller and Schiehlen [43], for the seismic excited coupled system

$$J(\nu) \ = \ \int_{-\infty}^{\infty} \vec{z}_S^T(\nu) \, \mathbf{S} \, \vec{z}_S(\nu) \, d\nu \ = \ 2\pi \, \vec{e}_g^T \mathbf{P} \, \vec{e}_g \quad \rightarrow \quad min, \tag{30}$$

and for the force excited coupled system,

$$J(\nu) \ = \ \int_{-\infty}^{\infty} \vec{z}_S^T(\nu) \, \mathbf{S} \, \vec{z}_S(\nu) \, d\nu \ = \ 2\pi \, \vec{b}_0^T \mathbf{P} \, \vec{b}_g \quad \rightarrow \quad min. \tag{31}$$

where the matrix **P** is the solution of the algebraic Lypanuov matrix equation,

$$(\mathbf{A} + \mathbf{B} \, \mathbf{R})^T \mathbf{P} \ + \ \mathbf{P} \, (\mathbf{A} + \mathbf{B} \, \mathbf{R}) \ = \ -\mathbf{S}. \tag{32}$$

The matrix **S** is a symmetric, positive semi-definite weighing matrix, which offers the opportunity to emphasize the importance of selected components of the state space vector. The matrix solution for **P** is numerically evaluated by means of the software MATLAB. The minimum search is best performed by the MATLAB optimization toolbox, substituting Den Hartog's modal design parameters for the SA-TLGDs as start values.

## 4. Numerical Studies on the Effectiveness of SA-TLGDs

### 4.1. SDOF-Wind Turbine with Single SA-TLGD Attached

The effectiveness of a single SA-TLGD regarding vibration reduction is demonstrated by considering a slender vibration-prone wind turbine with a flat gravity basement (Figure 4a) under severe wind-induced forcing by a time harmonic excitation force $F(t)$. The wind turbine is modeled as a clamped continuous Bernoulli–Euler beam with the height $h$ and the head mass $m_P$ on top (Figure 4b). Further $\rho A$ and $EI$ denote the tower mass per unit length as well as the bending stiffness. The mass $m_P$ is composed of the mass of the rotor blades, the hub, the nacelle including all internals, and the dead weight of the installed SA-TLGD. The first natural bending frequency $f_1$ with its corresponding mode shape $\phi_1$ is selected as most critical resonance frequency regarding wind-induced vortex shedding, and hence the continuous beam model with an infinite number of DOF can be reduced to a simple SDOF system. The position of the installed single SA-TLGD is assumed at hub height $h$. The damping ratio of the wind turbine consists of structural damping and aerodynamic damping. Light modal damping is assumed with $\zeta_S = 1.4\%$ according to [44] for steel towers.

The parameters chosen in this numerical study for the considered wind turbine are listed in Table 1.

**Table 1.** Parameters chosen for the considered wind turbine.

| Parameter of the Wind Turbine | Variable | Value | Unit |
|---|---|---|---|
| Hub height | $h$ | 60 | m |
| Tower diameter | $D$ | 5.5 | m |
| Tower mass per unit length | $\rho A$ | 3330 | kg/m |
| Bending stiffness | $EI$ | $2.9 \times 10^{11}$ | Nm$^2$ |
| Head mass | $m_P$ | 300,000 | kg |
| Modal damping ratio | $\zeta_S$ | 1.4 | % |

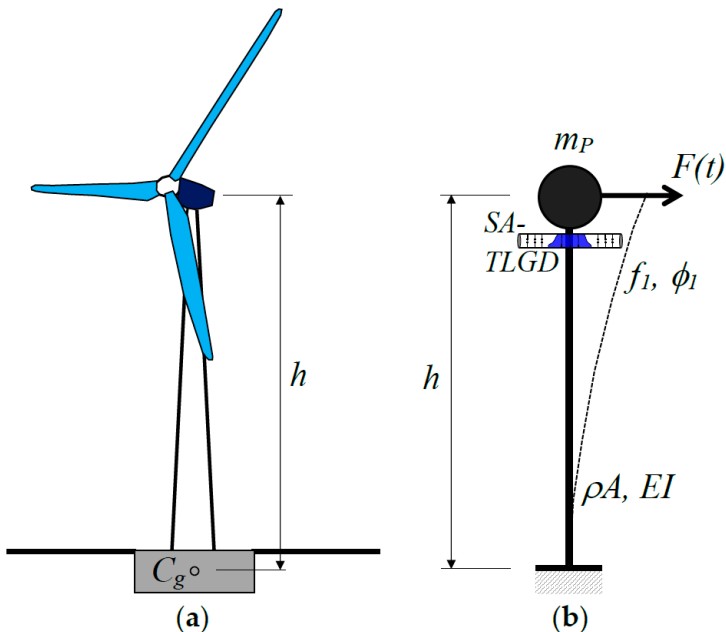

**Figure 4.** Wind turbine considered in the numerical study: (**a**) principial drawing of a wind turbine with a flat gravity basement, $C_g$ is the center of gravity; (**b**) mechanical model of the harmonic force excited wind turbine with a single SA-TLGD attached.

By applying the single-unit Ritz approach for the approximated solution of the dynamic deflection along the height of the considered wind turbine tower,

$$w(x,t) = q(t) \cdot \phi_1(x), \tag{33}$$

with the properly chosen shape function for the relevant fundamental vibration mode (Figure 2b), which fulfills the kinematic boundary conditions,

$$\phi_1(x) = 1 - \cos \frac{\pi x}{2h}, \tag{34}$$

yield the following formulas for the (kinetic equivalent) modal mass and stiffness of the SDOF-main system just from an energy comparison to the equivalent single-mass oscillator [39],

$$m_S = 0.227 \, \rho A h + m_P, \; k_S = \frac{\pi^4 EI}{32 \, h^3}. \tag{35}$$

Subsequently, the fundamental circular natural frequency of the SDOF-main system can be written as follows,

$$\omega_1 = \sqrt{\frac{k_S}{m_S}} = \frac{\pi^2}{h^2} \sqrt{\frac{EI}{32 \cdot (0.227 \rho A + m_P/h)}}. \tag{36}$$

Inserting the assumed wind turbine parameters listed in Table 1 into Equation (36) results in the circular natural frequency of the considered first vibration mode of the wind turbine to $\omega_1 = 3.44$ rad/s ($f_1 = \omega_1/2\pi = 0.55$ Hz). The comparison with the measured fundamental frequencies of existing onshore wind turbines shows a very good agreement [45,46]. While Equation (36) is simple to use, it is noted that it does not consider the flexibility of the foundation and stiffness softening effects due to the axial vertical load arising from the mass $m_P$. The influence of these in practice relevant effects on the resulting natural frequencies of wind turbines was studied in [45,46].

Optimal tuning of the single SA-TLGD with respect to the relevant fundamental frequency of the force excited SDOF-main system, $f_1 = 0.55$ Hz, is performed by the

application of Den Hartog's formulas (Section 3.1). The ratio of fluid mass to the modal mass of the SDOF-main system is chosen with $\mu = 1\%$. Inserting this mass ratio into Equation (22) and applying the correction formulas given in Equation (24) to account for the available light structural damping of the main system ($\zeta_S = 1.4\%$) renders the following optimal parameters for the SA-TLGD: $\omega_{A,opt} = 3.39$ rad/s ($f_{A,opt} = \omega_{A,opt}/2\pi = 0.54$ Hz) and $\zeta_{A,opt} = 6.28\%$. To realize the optimal natural frequency $f_{A,opt}$ in the practical application of the considered wind turbine, the dimensions of the SA-TLGD must be chosen appropriately. The optimal damping ratio $\zeta_{A,opt}$ is achieved by properly adjusting the diameter of the controllable orifices in the fluid stream. Table 2 lists both the calculated optimal parameters and the chosen dimensions of the SA-TLGD attached to the SDOF-wind turbine.

**Table 2.** Parameters and dimensions of the optimized SA-TLGD attached to the SDOF-wind turbine.

| Parameter of SA-TLGD | Variable | Value | Unit |
|---|---|---|---|
| Optimal natural frequency | $f_{A,opt}$ | 0.54 | Hz |
| Optimal damping ratio | $\zeta_{A,opt}$ | 6.28 | % |
| Mass ratio | $\mu = m_f/m_S$ | 1.0 | % |
| Fluid mass | $m_f$ | 3454 | kg |
| Fluid density | $\rho_f$ | 1000 | kg/m$^3$ |
| Fluid volume | $V_f$ | 3.454 | m$^3$ |
| Fluid horizontal length | $L_f$ | 4.40 | m |
| Polytropic coefficient | $n$ | 1.2 | - |
| Initial atmospheric pressure | $p_0$ | $10^5$ | Pa |
| Tube diameter | $d$ | 1.0 | m |
| Tube cross-sectional area | $A_T$ | 0.79 | m$^2$ |
| Gas spring length | $L_g$ | 4.75 | m |
| Initial gas volume | $V_0$ | 3.73 | m$^3$ |

Assuming the time harmonic horizontal excitation force with $F(t) = F_0\, e^{i\nu t}$, where $\nu$ is the excitation frequency, inserting the already chosen parameters for the considered wind turbine (Table 1) and for the optimally tuned SA-TLGD (Table 2) into Equation (17), yield the linearized matrix equations of the coupled system with now known mass, stiffness, and damping matrix elements. The steady state solutions for the tower head and fluid mass displacements $w(\nu)$ and $u(\nu)$ are determined in the frequency domain after inserting the exponential functions $w(\nu) = w_0\, e^{i\nu t}$ and $u(\nu) = u_0\, e^{i\nu t}$ as well as their derivatives into the matrix equation of the coupled system [7].

Figure 5 shows the gained results for the frequency response functions (FRF) of the SDOF-wind turbine with and without activated optimally tuned SA-TLGD. The force amplitude is chosen with $F_0 = 20,000$ N and the equivalent stiffness parameter of the wind turbine tower is defined by $k_S = m_S\,\omega_S^2 = 4.09$ MN/m, i.e., the static displacement of the tower head is given by $w_{st} = F_0/k_S = 0.0049$ m. It is seen that the activated optimally tuned SA-TLGD reduces the maximum horizontal displacement amplitude of the SDOF-wind turbine over the total frequency range of interest more than 70% (from 0.17 m without any vibration absorber to 0.05 m with activated SA-TLGD). The gained reduction in the maximum dynamic wind turbine head displacement corresponds to an effective structural damping ratio of $\zeta_{S,eff} = 4.9\%$ and this equates to an increase in the light structural damping of the wind turbine ($\zeta_S = 1.4\%$) of more than three times. Figure 6 illustrates the steady state response of the fluid mass displacement amplitude $u$ for the case of the activated SA-TLGD. It is indicated that the maximum fluid mass amplitude results in $u_{max} = 0.37$ m over the total frequency range of interest.

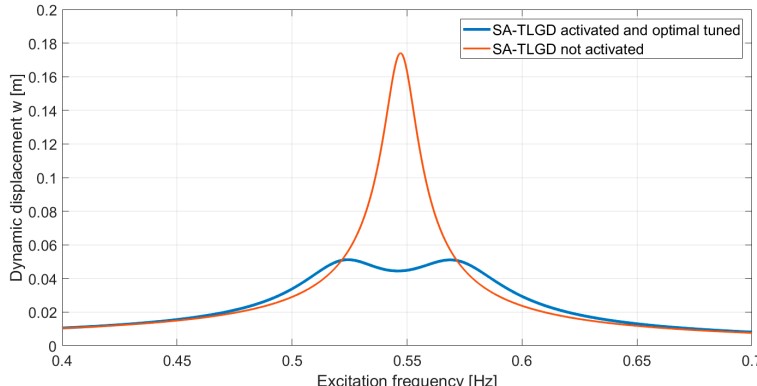

**Figure 5.** Frequency response functions (FRF) of the SDOF-wind turbine with and without activated optimally tuned SA-TLGD (steady state response of the horizontal tower head displacement *w*).

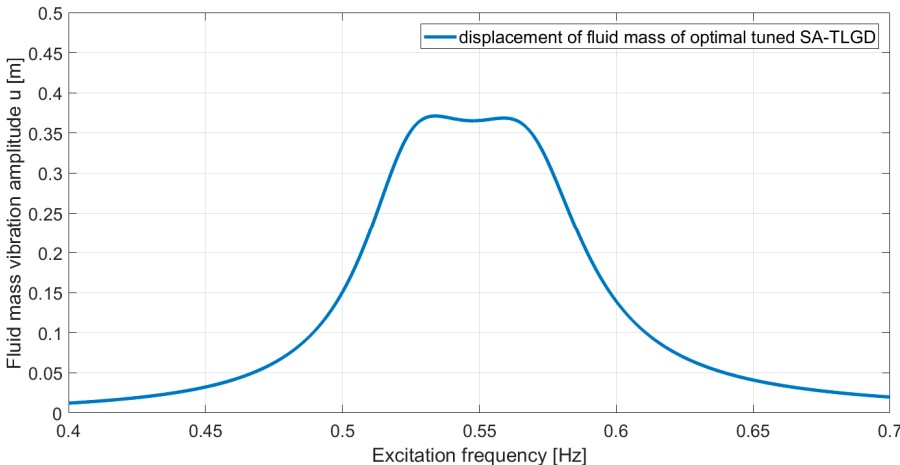

**Figure 6.** Frequency response function (FRF) of the activated optimally tuned SA-TLGD (steady state response of the fluid mass displacement amplitude *u*).

In the next step it is assumed that the natural frequency of the wind turbine $f_S = 0.55$ Hz varies $\pm 5\%$, i.e., $f_S^{+5\%} = 0.58$ Hz and $f_S^{-5\%} = 0.52$ Hz, and hence detuning of the SA-TLGD occurs, which obviously influences the achievable damping effect negatively. The assumed magnitude for the frequency shift is quite common in real structures and caused, for instance, by temperature effects and stiffness variations within the operating life of structures. Figure 7 clearly indicates that due to the assumed shifts in the fundamental frequency the maximum head displacement *w* increases from the minimum value 0.05 m for the optimally tuned SA-TLGD, to around 0.08 m.

To account for the expected frequency shifts that occur in real structures and that lead to a detrimental detuning of the installed vibration absorber, the SA-TLGD is equipped with a total of three gas volumes $V_{01}$, $V_{02}$, and $V_{03}$ at each side of the tube (cf. illustration in Figure 1). Beside the optimal eigenfrequency of the SA-TLGD, $f_{A,opt} = 0.54$ Hz listed in Table 2, that corresponds to the unchanged natural frequency $f_S = 0.55$ Hz, the optimal eigenfrequencies that correspond to the shifted natural frequencies yield from Equations (22) and (24) to $f_{A,opt}^{-5\%} = 0.51$ Hz and $f_{A,opt}^{+5\%} = 0.57$ Hz. Subsequently, the required sizes of the gas volumes are calculated, based on the chosen SA-TLGD dimensions listed in Table 2, from Equation (9) to $V_{01} = 3.34$ m$^3$ ($L_{g1} = 4.26$ m), $V_{02} = 0.39$ m$^3$ ($L_{g2} = 0.49$ m), and $V_{03} = 0.35$ m$^3$ ($L_{g3} = 0.45$ m) to achieve the corresponding vibration frequency of the SA-TLGD. It is noted that the separated gas chambers are connected via controllable valves (cf. again Figure 1) and that, for instance, the required gas volume to achieve the lowest eigenfrequency $f_{A,opt}^{-5\%} = 0.51$ Hz relates to the gas volume size of $V_{01} + V_{02} + V_{03} = 4.08$ m$^3$.

In practical applications, the air chamber volumes $V_{0i}$ can be redesigned with a smaller or larger cross-sectional area than $A_T$, i.e., basically the geometric shape of the air chambers is arbitrary and therefore adjustable to the available installation space of the structure. Obviously, the cross-sectional area of the tube $A_T$ must be kept constant at least up to the maximum dynamic vibration amplitude $u = U_{max}$ of the fluid mass.

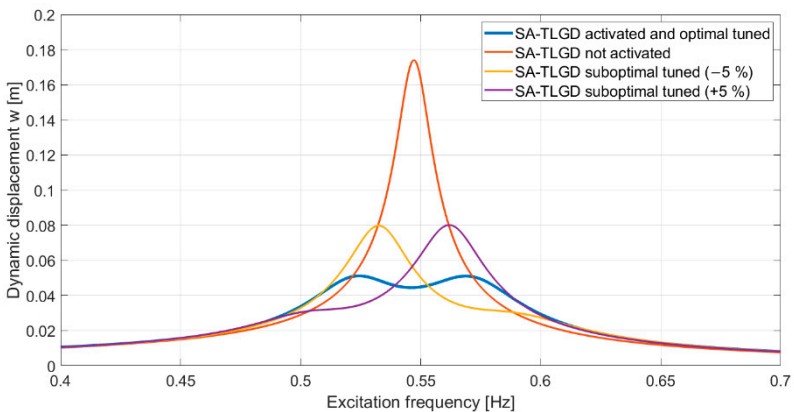

**Figure 7.** Frequency response functions (FRF) of the SDOF-wind turbine with sub-optimal tuned SA-TLGD (frequency shift of the wind turbine fundamental mode $f_S$ = 0.55 Hz to $f_S^{+5\%}$ = 0.58 Hz and $f_S^{-5\%}$ = 0.52 Hz assumed).

### 4.2. MDOF-Shear Frame Structure with Multiple SA-TLGDs Attached

The effectiveness of multiple SA-TLGDs in vibration reduction is demonstrated for a plane three-DOF-shear frame structure under horizontal force excitation, e.g., caused by wind gusts. Based on a benchmark definition paper from Spencer et al. [47], a scale model of the original structure was built at the Multidisciplinary Center for Earthquake Engineering Research (MCEER) at Buffalo. In the following numerical study, the considered force excited MDOF-shear frame model with a total mass of 2943 kg and a total height of 2.55 m is equipped with two SA-TLGDs in parallel connection on the 3rd floor (see Figure 8). The optimization procedure of multiple vibration absorbers that are supposed to be tuned to the most critical resonance frequency of the structure, requires fine-tuning in the state space (see Section 3.2). Modal (SDOF) tuning as discussed in Section 3.1 is performed in a first step.

The mass- and stiffness matrix of the scale MDOF-shear frame model are provided in [48],

$$\mathbf{M} = \begin{bmatrix} 981 & 0 & 0 \\ 0 & 981 & 0 \\ 0 & 0 & 981 \end{bmatrix} [\text{kg}]. \ \mathbf{K} = \begin{bmatrix} 650.3 & -183.4 & 33.2 \\ -183.4 & 574.7 & -148.9 \\ 33.2 & -148.9 & 387.2 \end{bmatrix} [\text{N/m}], \quad (37)$$

and the orthonormalized eigenvectors and well-separated undamped natural frequencies are yielded after solving the eigen value problem [39],

$$\vec{\phi}_1 = \begin{bmatrix} 0.2015 \\ 0.5472 \\ 0.8123 \end{bmatrix}, \ \vec{\phi}_2 = \begin{bmatrix} 0.6782 \\ 0.5204 \\ -0.5189 \end{bmatrix}, \ \vec{\phi}_3 = \begin{bmatrix} -0.7067 \\ 0.6555 \\ -0.2662 \end{bmatrix},$$

$$f_{S1} = 2.38 \text{ Hz}, \ f_{S2} = 7.44 \text{ Hz}, \ f_{S3} = 12.29 \text{ Hz.} \tag{38}$$

The light modal damping ratios are set to $\zeta_{S1}$ = 1%, $\zeta_{S2}$ = 2%, and $\zeta_{S3}$ = 3%, respectively. Both attached SA-TLGD are tuned with respect to the most critical fundamental natural frequency $f_{S1}$ = 2.38 Hz choosing the mass ratio $\mu$ = 4% with respect to the moving modal mass $M_1$ = 1531 kg of the fundamental vibration mode. Thereby, the modal mass is determined by calculating the kinetic energy of the vibrating MDOF-main system in the first mode and comparing it with the kinetic energy of an equivalent single-mass oscillator [39].

With $\mu$ = 4%, the fluid mass of each attached SA-TLGD results in $m_{f1} = m_{f2}$ = 30 kg, i.e., the total fluid mass is $m_{f,total}$ = 60 kg.

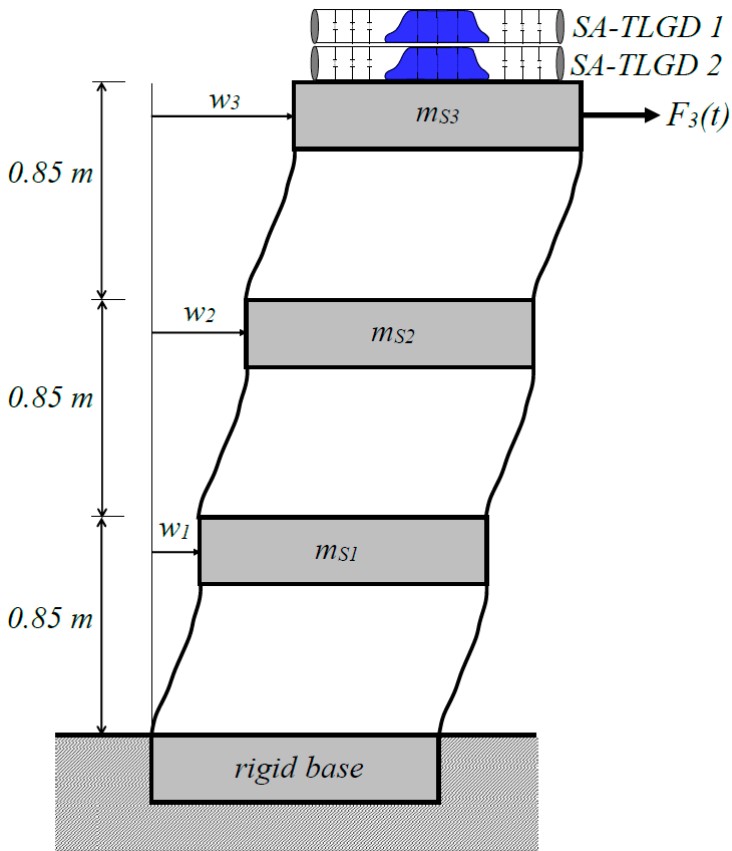

**Figure 8.** Scaled three-DOF-shear frame structure under horizontal force excitation $F_3(t)$ with two SA-TLGDs in parallel connection attached to the 3rd floor.

In a first step, the optimal absorber eigenfrequencies $f_{A1,opt}$, $f_{A2,opt}$, and optimal linearized viscous damping ratios $\zeta_{A1,opt}$, $\zeta_{A2,opt}$ of the two attached SA-TLGDs are determined by the application of Den Hartog's formulas. Inserting $\mu$ = 4% into Equation (22) yields identical optimal design parameters for both SA-TLGDs: $f_{A1,opt} = f_{A2,opt}$ = 2.29 Hz, $\zeta_{A1,opt} = \zeta_{A2,opt}$ = 12.01%. Improvements in their performance are achieved by minimizing the frequency domain-based quadratic performance index $J(\nu)$ in the state space representation (see Section 3.2). The state vector of the MDOF-shear frame structure, $\vec{z}_S = [w_1, w_2, w_3, \dot{w}_1, \dot{w}_2, \dot{w}_3]^T$, to be substituted in Equation (31), does not contain the SA-TLGD quantities explicitly. However, the vibration damping effects of the attached SA-TLGDs are hidden in the system's dynamics, Equation (29), and thus in the structural response state vector $\vec{z}_S$. The relevant system's matrices **A**, **B**, and **R** are defined in Equations (26) and (27) and they contain both, the MDOF-main system's parameters and the quantities of the two attached SA-TLGDs.

Having chosen the weighing matrix, $S = diag[10, 10, 10, 1, 1, 1]$ in Equation (32), the numerical minimization of the performance index $J(\nu)$ is started with Den Hartog's modal tuning parameters as initial values. Calling the command *fminsearch* within the MATLAB optimization toolbox renders the final improved optimal tuning parameters for the two attached SA-TLGDs, significantly changed to: $f_{A1,opt}$ = 2.18 Hz, $f_{A2,opt}$ = 2.49 Hz, and $\zeta_{A1,opt}$ = 6.08%, $\zeta_{A2,opt}$ = 6.72%. It is noticed that the eigenfrequencies $f_{A1,opt}$ and $f_{A2,opt}$ are significantly smaller and larger, respectively, than the considered fundamental natural frequency of the MDOF-shear frame structure $f_{S1}$ = 2.38 Hz. This gained properties regarding the eigenfrequencies of the two SA-TLGDs to increase the robustness of the

damping absorbers with respect to the expected changes of the fundamental natural frequency during the operating life. Based on the determined optimal tuning parameters the dimensions of the two SA-TLGDs must be chosen with respect to the available space at the 3rd floor of the MDOF-shear frame structure. The optimal damping ratios $\zeta_{Ai,opt}$ are achieved by properly adjusting the diameters of the controllable orifices in the fluid stream. The calculated optimal parameters and chosen dimensions of the installed two SA-TLGDs are listed in Table 3.

Assuming a time harmonic excitation force $F_3(t)$ with the force amplitude $F_{03} = 1000$ N acting on the 3rd floor of the MDOF-shear frame structure with the two optimally tuned SA-TLGDs attached (Figure 8) results in the frequency response function (FRF) for the horizontal floor displacement $w_3(t)$ illustrated in Figure 9. The two installed, activated, and optimally tuned SA-TLGDs reduce the maximum floor displacement from 0.143 m to just 0.017 m around the most critical fundamental mode of the MDOF-shear frame structure, i.e., the percentage of vibration reduction is almost 90%. In the case of the activated SA-TLGDs the FRF of the floor displacement indicates three peaks in the vicinity of the fundamental mode and obviously, these peaks arise due to the determined different optimal tuning parameters for both attached SA-TLGDs. Because the two SA-TLGDs are optimally tuned to the fundamental frequency of the MDOF-shear frame structure $f_{S1} = 2.38$ Hz, the higher modes $f_{S2} = 7.44$ Hz and $f_{S3} = 12.29$ Hz are not affected by the installed vibration absorbers. The modal stiffness parameter of the fundamental mode is defined by $K_1 = M_1 \omega_{S1}^2 = 4.09$ MN/m, i.e., the static displacement of the 3rd floor results in $w_{3,st} = F_{03}/K_1 = 0.0029$ m. Hence, the gained reduction in the maximum floor displacement $w_3$ corresponds to an effective structural damping ratio of $\zeta_{S1,eff} = 8.6\%$ and this equates to an increase in the assumed light structural damping for the fundamental mode ($\zeta_{S1} = 1\%$) of almost nine times.

**Table 3.** Parameters and dimensions of the two optimized SA-TLGDs attached to the 3rd floor of the MDOF-shear frame structure.

| Parameter of SA-TLGD | Variable | TLGD 1 | TLGD 2 | Unit |
|---|---|---|---|---|
| Optimal natural frequency | $f_{Ai,opt}$ | 2.18 | 2.49 | Hz |
| Optimal damping ratio | $\zeta_{Ai,opt}$ | 6.08 | 6.72 | % |
| Total fluid mass | $m_{f,total}$ | 60 | | kg |
| Total mass ratio | $\mu_{total} = m_{f,total}/M_1$ | 4.0 | | % |
| Mass ratio of each SA-TLGD | $\mu_i = m_{fi}/M_1$ | 2.0 | | % |
| Fluid mass of each SA-TLGD | $m_{fi}$ | 30 | | kg |
| Fluid density | $\rho_f$ | 1000 | | kg/m$^3$ |
| Fluid volume | $V_f$ | 0.031 | | m$^3$ |
| Fluid horizontal length | $L_f$ | 0.624 | | m |
| Polytropic coefficient | $n$ | 1.2 | | - |
| Initial atmospheric pressure | $p_0$ | 10$^5$ | | Pa |
| Tube diameter | $d$ | 0.25 | | m |
| Tube cross-sectional area | $A_T$ | 0.049 | | m$^2$ |
| Gas spring length | $L_g$ | 2.05 | 1.57 | m |
| Initial gas volume | $V_0$ | 0.1007 | 0.0772 | m$^3$ |

To investigate the detrimental influence of an expected frequency shift of the MDOF-shear frame fundamental frequency $f_{S1} = 2.38$ Hz during the operating life on the vibration damping effectiveness gained through the two optimally tuned SA-TLGDs the following assumptions are made: $f_{S1}^{+5\%} = 2.50$ Hz and $f_{S1}^{-5\%} = 2.26$ Hz. The FRF of the detuned force excited MDOF-shear frame structure with the two sub-optimal tuned SA-TLGDs in parallel connection is illustrated in Figure 10. Note that the FRF of the activated and optimally tuned SA-TLGD is also illustrated in Figure 10. It is seen that due to the assumed frequency shift of the fundamental mode of the MDOF-shear frame structure the maximum floor displacement $w_3$ increases from the minimum value 0.018 m for the two optimally tuned SA-TLGDs, to around 0.023 m.

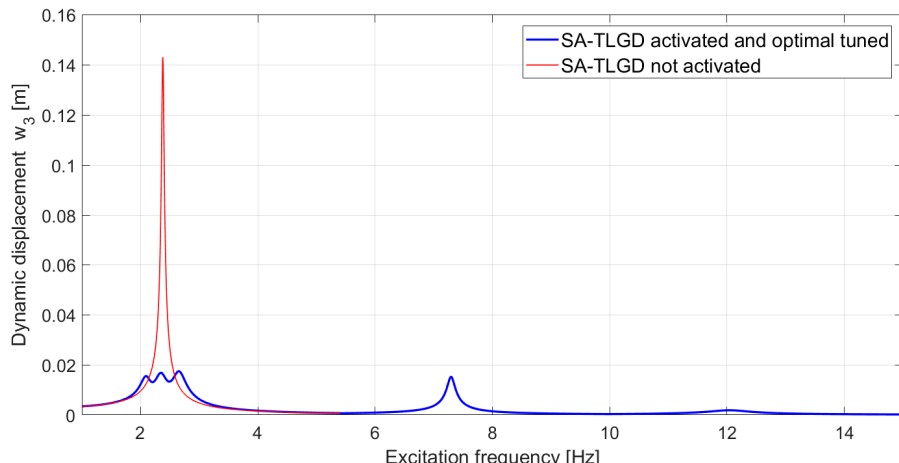

**Figure 9.** Frequency response functions (FRF) of the force excited MDOF-shear frame structure with and without the two activated optimally tuned SA-TLGDs (steady state response of floor displacement $w_3$).

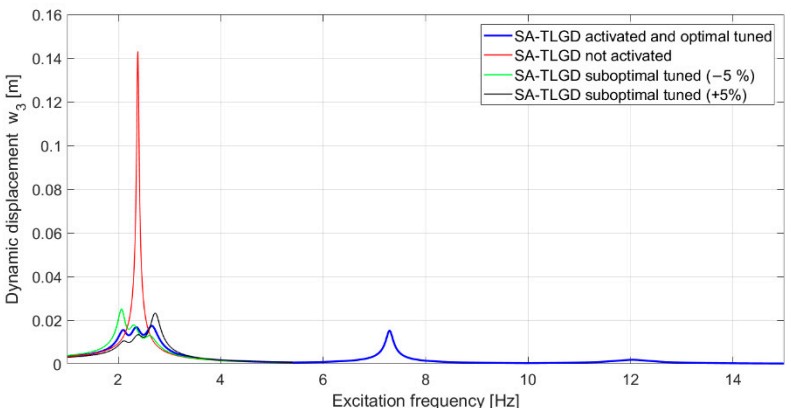

**Figure 10.** Frequency response functions (FRF) of the force excited MDOF-shear frame structure with two sub-optimal tuned SA-TLGDs in parallel connection (steady state response of floor displacement $w_3$; frequency shift of the fundamental natural frequency $f_{S1} = 2.38$ Hz to $f_{S1}^{+5\%} = 2.50$ Hz and $f_{S1}^{-5\%} = 2.26$ Hz assumed).

To account for the above-mentioned shift of the fundamental frequency of the MDOF-main system, the two SA-TLGDs are equipped with a total of three gas volumes $V_{01}$, $V_{02}$, and $V_{03}$ at each side of the tube. The optimal eigenfrequencies of the SA-TLGDs that correspond to the shifted natural frequencies yield again by minimizing the frequency domain based quadratic performance index $J(\nu)$ in the state space representation (see Section 3.2): $f_{A1,opt} = 2.07$ Hz, $f_{A2,opt} = 2.36$ Hz in the case of $f_{S1}^{-5\%} = 2.26$ Hz and $f_{A1,opt} = 2.29$ Hz, $f_{A2,opt} = 2.61$ Hz in the case of $f_{S1}^{+5\%} = 2.50$ Hz. The linearized viscous damping ratios of the two SA-TLGDs remain unchanged. Subsequently, the required sizes of the gas volumes for the two installed SA-TLGDs, based on the chosen absorber dimensions listed in Table 3, from Equation (9) are: $V_{01,A1} = 0.0912$ m$^3$ ($L_{g1,A1} = 1.86$ m), $V_{02,A1} = 0.0095$ m$^3$ ($L_{g2,A1} = 0.19$ m), $V_{03,A1} = 0.0110$ m$^3$ ($L_{g3,A1} = 0.22$ m) for the first installed SA-TLGD and $V_{01,A2} = 0.0702$ m$^3$ ($L_{g1,A2} = 1.43$ m), $V_{02,A2} = 0.0069$ m$^3$ ($L_{g2,A2} = 0.14$ m), $V_{03,A2} = 0.0087$ m$^3$ ($L_{g3,A2} = 0.18$ m) for the second installed SA-TLGD.

The separated gas chambers are connected via controllable valves (cf. again Figure 1). To achieve the optimal absorber tuning frequency the appropriate number of valves must be opened or closed. In practical applications, the air chamber volumes $V_{0i}$ can be redesigned with a smaller or larger cross-sectional area than $A_T$, i.e., basically the geometric shape of the air chambers is arbitrary and therefore adjustable to the available installation space of

the structure. Obviously, the cross-sectional area of the tube $A_T$ must be kept constant at least up to the maximum dynamic vibration amplitude $u = U_{max}$ of the fluid mass.

## 5. Conclusions

This paper introduced a novel single tube semi-active tuned liquid gas damper (SA-TLGD) for suppressing horizontal vibrations of tower-like structures. The special feature of the presented SA-TLGD is its lack of any vertical tube sections, which lead to a great advantage regarding the required installation space in slender vibration-prone structures, e.g., wind turbines. The horizontal orientated single tube is partially filled with a fluid and sealed at both ends. A large deformable elastic membrane with neglectable stiffness is used as the interface between fluid and air, and the resulting gas spring provides the restoring force and frequency tuning parameter, respectively.

The equations of motion were derived for both a SDOF-main system with a single SA-TLGD and a MDOF-main system with multiple SA-TLGDs attached. Modal tuning of the single SA-TLGD attached to a SDOF-main system was presented by the application of Den Hartog´s formulas considering a harmonic force and seismic excitation. Optimal tuning of multiple SA-TLGDs attached to a MDOF-main system was achieved by minimizing the frequency domain-based quadratic performance index in state space representation.

It was shown that the adjustment of the SA-TLGDs vibration frequency is simply achieved by separating the bulk gas volume $V_0$ at the left and right tube sections into a series of gas chambers $V_{0i}$ all connected via controllable valves. Depending on the desired optimal vibrating frequency a specific size of gas volume is initiated through the utilized control software, which opens or closes the appropriate number of valves. In addition, the magnitude of the fluid damping is properly adjusted by varying the diameter of several controllable orifices that are built into the fluid stream.

The achievable damping effectiveness of the introduced SA-TLGDs were evaluated considering two different application examples. The first example was a SDOF-wind turbine with a single optimally tuned SA-TLGD attached. In this example, the vibration absorber was optimally tuned to the most critical fundamental mode of the force excited SDOF-main system and a frequency shift of $\pm 5\%$ of the main systems' fundamental frequency was assumed. The mass ratio was chosen with $\mu = 1\%$. It was shown that the achievable reduction in the maximum tower head displacement resulted as almost 70% and that the assumed light structural damping ratio of $\zeta_S = 1.4\%$ could be increased to the effective damping ratio of $\zeta_{S,eff} = 4.9\%$ by the installed optimally tuned SA-TLGD. To avoid detuning due to the assumed frequency shift the SA-TLGD was equipped with a total of three gas volumes $V_{01}$, $V_{02}$, and $V_{03}$ at each side of the sealed tube and optimal tuning was achieved by setting the appropriate size of the gas volume via controllable valves.

The second example was a scaled MDOF-shear frame structure with two optimally tuned SA-TLGDs in parallel connection installed on the top floor of the structure. The mass ratio was chosen with $\mu = 4\%$ (includes the total fluid mass of both installed SA-TLGDs). The two vibration absorbers were tuned to the most critical fundamental vibration mode of the MDOF-main system, and it was shown that a vibration reduction in the maximum horizontal 3rd floor displacement of almost 90% could be achieved. The assumed light structural damping ratio of $\zeta_{S1} = 1\%$ of the fundamental mode could be increased to the effective damping ratio $\zeta_{S1,eff} = 8.6\%$ by the installed optimally tuned SA-TLGDs.

It is concluded that the presented SA-TLGDs achieve a high reduction in the maximum force vibration amplitudes of tower-like structures and that its semi-active functionality enables the possibility of re-adjustment any time during the operation of the damper. Hence, SA-TLGDs can best account for the expected frequency shifts that occur in real structures and, thus, provide the favored optimal performance over the total operation life of vibration-prone structures. Furthermore, the lack of any vertical tube sections of the presented SA-TLGD makes it easier to implement the vibration absorber into slender vibration-prone structures and, thus, opens a whole new field of possible applications.

In the next step of the research work we plan to set up a laboratory tests with the SA-TLGD on a small scale using a uniaxial shaking table and to study the achievable effectiveness as well as the practical implementation of the presented vibration absorber.

**Author Contributions:** Conceptualization, M.R.; introduction, M.R. and J.S.; mechanical model, M.R.; optimization, M.R. and J.S.; numerical studies, M.R.; writing—original draft preparation, M.R.; writing—review and editing, J.S.; visualization, M.R.; supervision, M.R.; project administration, M.R.; funding acquisition, M.R. All authors have read and agreed to the published version of the manuscript.

**Funding:** This research was funded by the Austrian Research Promotion Agency (Österreichische Forschungsförderungsgesellschaft, FFG) under the grant number 887996.

**Institutional Review Board Statement:** Not applicable.

**Informed Consent Statement:** Not applicable.

**Data Availability Statement:** The data presented in this article is available on request from the corresponding author.

**Acknowledgments:** The authors appreciate the financial support of this study by the Austrian Research Promotion Agency (FFG).

**Conflicts of Interest:** The authors declare no conflict of interest. The funders had no role in the design of the study; in the collection, analyses, or interpretation of data; in the writing of the manuscript, or in the decision to publish the results.

## Abbreviations

The following abbreviations are used in this manuscript:

| | |
|---|---|
| FRF | Frequency Response Function |
| MDOF | Multiple Degrees Of Freedom |
| MR-TLCD | Magneto-Rheological Tuned Liquid Column Damper |
| MW | Megawatt |
| SA-TLGD | Semi-Active Tuned Liquid Gas Damper |
| SDOF | Single Degree Of Freedom |
| TMD | Tuned Mass Damper |
| TLCD | Tuned Liquid Column Damper |
| TLCGD | Tuned Liquid Column Gas Damper |
| TLD | Tuned Liquid Damper |
| TSD | Tuned Sloshing Damper |

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
