# Peer review of "A Novel Single Tube Semi-Active Tuned Liquid Gas Damper for Suppressing Horizontal Vibrations of Tower-like Structures"

_applsci, doi:10.3390/app12073301_

Round 1
Reviewer 1 Report
The manuscript presented a single tube semi-active tuned liquid gas damper (SA-TLGD) to reduce the vibrations of tower-like structures. This manuscript is overall well structured and written, of which the topic is of interest and relevance and fits the journal. The reviewer has the following comments that the authors have to address before the manuscript can be considered for publication.
(1) However, considering the used over-simple and ideal analytical model with no experimental test, the novelty and contribution of this manuscript should be highlighted and explained in more detail.
(2) The finite element simulation or experimental test are suggested to validate the accuracy of the employed analytical model. In the analytical model with the linearized method, the SA-TLGD is identical to the TMD, which may underestimate the nonlinear behavior and benefit of this damper.
(3) The analyzed damper is called as ‘semi-active tuned liquid gas damper (SA-TLGD)’, which can be adjusted by the microcontroller (MC). However, the detailed time variable control method and practical implementation are missed.
(4) The SA-TLGD is optimized with a relatively high damping ratio, which is difficult to realize in the application. Please explain and add the rational implementation method.
(5) Apart from the well-known calculation formulas presented in this reporting process, the authors should further explain and illustrate the theoretical contributions of this study in terms of the SPECIAL design methods and performance improvements for existing buildings.
(6) Ref [34] is in German. A substitute in English is recommended. Within Introduction, please add and discuss the following studies dealing with enhanced tuned liquid dampers, such as:
[1] A compliant tuned liquid damper for controlling seismic vibration of short period structures, Mechanical Systems and Signal Processing, Volume 132, 1 October 2019, Pages 405-428.
[2] A tuned liquid inerter system for vibration control. International Journal of Mechanical Sciences, 2019, 164, 105171.
[3] A tuned liquid mass damper implemented in a deep liquid storage tank for seismic vibration control of short period structures. The Structural Design of Tall and Special Buildings, e1928.
Reviewer 2 Report
The article proposes an additional dissipation technique for controlling the vibrations of tall structures with a mainly linear shape. The idea is interesting and well consistent with the topics of the journal. The main novelty should be the fact that the proposed tuned liquid damper has a horizontal axis and this would simplify its use.
This reviewer appreciated the work done by the authors, but asks them for clarification and suggestions for improvement of the article before it can be accepted for publication:
- when you deal with "semi-active" systems, it is necessary to place significant emphasis on the definition of the "control logic", that is the algorithm with which, due to the particular characteristics of the dynamic input or the structural response, it is automatically and real-time optimized device for higher vibration suppression. This part is not covered enough in the article. It almost seems like the design of a passive system, and this confuses the reader. It is necessary to clarify which are the parameters (input load or structural demand) that are monitored and how these are used for the optimal calibration of the device.
- Still with regard to "semi-active" techniques, delays in the control chain (acquisition, processing, decision, device calibration) or the speed of adaptation of the system generally play an important role. Any comment on this is also expected.
- Some comments should be added regarding the installation methods (even not detailed, but at least preliminary) for the device and the resistance it must have to resist gravitational and non-gravitational loads.
- Does the tube have a fixed direction or can it rotate to function in different planes?
- Minor comments are about some ambiguous comments: e.g. line 338 page 10, authors are talking about SDOF systems then about "floor displacement"; the latter reference should be better explained or replaced, it seems hardly compatible with a SDOF system.
- References are well done, however it should be enriched with references particularly dedicated to control with TLD and / or SA techniques to tower-like structures. Some examples:
- Caterino, N., Georgakis, C.T., Trinchillo, F., and Occhiuzzi, A. (2014). A Semi-Active Con-trol System for Wind Turbines. In: Luo N, Vidal Y and Acho L (eds) Wind Turbine Con-trol and Monitoring. Advances in Industrial Control, Springer International Publishing Switzerland, ISBN 978-3-319-08412-1.
- Sarkar S, Chakraborty A. Optimal design of semiactive MR-TLCD for along-wind vibration control of horizontal axis wind turbine tower. Struct Control Health Monit 2018;25(2):1–18.
- H. R. Karimi, M. Zapateiro, and N. Luo, “Semiactive vibration control of offshore wind turbine towers with tuned liquid column dampers using H∞ output feedback control,” Proceedings of the IEEE International Conference on Control Applications, pp. 2245-2249.
Reviewer 3 Report
The paper “A novel single tube semi-active tuned liquid gas damper for suppressing horizontal vibrations of tower-like structures” reports a very interesting work about the application of a new SA-TLGD (semi-active tuned liquid gas damper) useful to control the vibration of tower-like structures under different type of dynamic loads. The original aspects of the paper are clearly described in the text where both theory and application of the new device proposed are discussed in detail. In particular, Sections 2 and 3 report the mechanical model and the optimal tuning of the SA-TLGDs and in Section 4 the application of the device on a wind turbine and on a MDOF system highlighting its effectiveness. For these reason it is opinion of this reviewer that the manuscript can be considered for the publication in Applied Scinces Journals after the following minor improvements:
- line 36-38 consider as reported in 10.1016/j.engstruct.2018.12.092 10.1016/j.engstruct.2016.06.006 https://doi.org/10.1016/j.istruc.2021.02.053
- Insert the dimensions in Figure 4a
- Table 1: Why the modal damping ratio is taken equal to 1.4%? Is it not a low value for steel wind turbines?
Round 2
Reviewer 1 Report
The manuscript has been revised according to the reviewer's comments. The reviewer does not have any further comments.